# The In Vivo Quantitative Assessment of the Effectiveness of Low-Dose Photodynamic Therapy on Wound Healing Using Optical Coherence Tomography

**DOI:** 10.3390/pharmaceutics14020399

**Published:** 2022-02-11

**Authors:** Hala Zuhayri, Viktor V. Nikolaev, Tatiana B. Lepekhina, Ekaterina A. Sandykova, Natalya A. Krivova, Yury V. Kistenev

**Affiliations:** Laboratory of Laser Molecular Imaging and Machine Learning, Tomsk State University, Lenin Ave. 36, 634050 Tomsk, Russia; zuhayri.n.hala@gmail.com (H.Z.); vik-nikol@bk.ru (V.V.N.); tatiana_lepekhina@mail.ru (T.B.L.); katrin_nemchenko@mail.ru (E.A.S.); nakri@res.tsu.ru (N.A.K.)

**Keywords:** low dose photodynamic therapy, wound healing, 5-aminolevulinic acid, methylene blue, optical coherence tomography

## Abstract

The effect of low-dose photodynamic therapy on in vivo wound healing was investigated using optical coherence tomography. This work aims to develop an approach to quantitative assessment of the wound’s state during wound healing including the effect of low-dose photodynamic therapy using topical application of two different photosensitizers, 5-aminolevulinic acid and methylene blue, and two laser doses of 1 J/cm^2^ and 4 J/cm^2^. It was concluded that the laser dose of 4 J/cm^2^ was better compared to 1 J/cm^2^ and allowed the wound healing process to accelerate.

## 1. Introduction

According to the World Health Organization (WHO), burn wounds result in approximately 180,000 deaths every year and nearly 11 million injuries that require medical treatment worldwide [1]. Cutaneous wounds are widespread and differentiated into acute and chronic wounds [2].

Wound healing is a complex physiological process at the cellular and molecular levels including the extracellular matrix synthesis, the replacement of type III collagen with type I collagen, and scar tissue formation [3,4,5,6]. These processes are divided into four overlapping stages: coagulation (hemostasis), inflammation, proliferation, and remodeling [7,8]. Some underlying diseases affect the wound healing process including peripheral arterial and venous disease or diabetes; acute wounds may have impaired healing, which can lead to a chronic stage [9,10,11]. In developed countries, 1–6% of the population suffers from chronic wounds [12,13,14].

It is known that a low dose photo process with photoactive compounds promotes the healing of skin diseases and leads to results in rejuvenation and wound healing [15,16]. Low-dose photodynamic therapy (LDPDT) is widely used to treat skin diseases and wound healing where it reduces the treatment time, accelerates tissue repair, and promotes healing [17,18]. The method is based on using a photosensitizer (PS), which accumulates in tissues, followed by irradiation of the tissue with a light source with an appropriate wavelength. The latter causes the formation of reactive oxygen species (ROS) [19,20]. Low concentrations of ROS can trigger cell repair processes including proliferation and offer promising treatments to accelerate healing. Different PSs have been studied in the wound healing process, which has a relevant role in ensuring PDT effectiveness in skin wound healing [21] such as 5-aminolevulinic acid (5-ALA) and methylene blue (MB) [22,23,24,25]. MB is a popular PS among the phenothiazinium derivatives that have attracted the attention of different research groups working and achieving good results in wound healing [25,26,27]. Recently, MB was shown to have an antioxidant role [28]. Additionally, 5-ALA is among the most effective photosensitizers and is widely used to present a better achievement concerning wound healing [17,22,29].

The Arndt–Schultz Law is an appropriate model to demonstrate that low levels of light have a better effect in wound healing than higher levels, which may have an inhibitory or cytotoxic effect [30,31]. Hawkins and Abrahamse studied the behavior in vitro of human skin fibroblasts using different irradiation doses of 0.5, 2.5, 5, 10, and 16 J/cm^2^. They demonstrated that higher laser doses (10 and 16 J/cm^2^) resulted in increased cellular damage as well as decreased cell viability and proliferation [32]. Results for different energy doses were described for 4 J/cm^2^ [33] and for 1 J/cm^2^ and 2 J/cm^2^ [34]. Basso et al. demonstrated that irradiation of cultured human gingival fibroblasts with energy doses of 0.5 and 3 J/cm^2^ resulted in a significant increase in cellular metabolism compared with the non-irradiated control group and the cells irradiated with higher energy doses of 5 and 7 J/cm^2^ [35]. The most significant biological effects were seen with predominant dose values (i.e., up to 5 J/cm^2^), which were within the Arndt–Schultz curve [36].

Traditionally, wounds have been observed invasively with a histochemical assessment of the biopsies [3,8]. Visual observation is a common tool for wound assessment. Additionally, clinical wound evaluation is a widely used and the least expensive method of assessing wound depth. This method relies on a subjective evaluation of the external features of the wound such as wound appearance, capillary refill, and burn wound sensibility to touch and pinprick, providing diagnostic accuracy at the level of 60–75% [37]. These methods are not quantitative and can lead to additional tissue damage and impair healing. Accordingly, the development of noninvasive and accurate methods of wound analysis is relevant.

Optical coherence tomography (OCT) is a noninvasive 3D imaging method of biological tissues with a spatial resolution of 5–10 µm and a penetration depth of 1–2 mm [38,39]. Epidermal thickness is a critical parameter for assessing epithelialization during wound healing [40,41].

OCT could detect essential morphological changes during wound healing (e.g., epidermis, dermis, adipose tissue, and granulation) that was based primarily on their backscattering characteristics [42,43,44]. The use of polarization-sensitive OCT revealed higher birefringence in scars compared to healthy skin [45]. OCT-based angiography provides in vivo, three-dimensional vascular information by using flowing red blood cells as intrinsic contrast agents, allowing visualization of functional vessel networks within microcirculatory tissue beds non-invasively, without needing dye injection [46].

This work aims to develop a method for quantitative in vivo evaluation of wounds using OCT during the wound healing process including a quantitative assessment of the effect of LDPDT using the topical application of two different photosensitizers (5-ALA and MB) and two laser doses of 1 J/cm^2^ and 4 J/cm^2^.

## 2. Materials and Methods

### 2.1. Wound Model Protocol

This study used 15 male laboratory CD1 mice, weighing 25–30 g and aged 6–7 weeks, obtained from the Department of Experimental Biological Models of the Research Institute of Pharmacology, TSC SB RAMS. Before the experiment, the mice were kept seven days in the standard conditions of a conventional vivarium with free access to water and food, and a 12/12 light regime, in a ventilated room at a temperature of 20 ± 2 °C and a humidity of 60%. The experimental protocol of this research was approved by the Bioethical Committee of Tomsk State University (Protocol No. 4, 10.02.2021), registration No. 6.

The mice were anesthetized by isoflurane using the Ugo Basile gas anesthesia system, where the mice were put in a glass chamber connected to isoflurane (Figure 1). The wound area was prepared through depilation using Veet cream (made in France), rinsed with saline solution, and sterilized using chlorhexidine 20%. A full-thickness cutaneous wound (diameter 5 mm) was formed by cutting out a whole layer skin flap with scissors on both of the hind paws of each animal under isoflurane anesthesia. The experiment was performed in a time-lapsed schedule for the wound aging on days 1, 3, 7, and 14. Additionally, the day of wound formation was defined as day 0.

### 2.2. Low Dose Photodynamic Therapy Protocol

Both of the photosensitizers 5-ALA 20% and MB 0.01% in saline solution were topically administered directly on the wound; after 30 min, the irradiation was started by an AlGalnP laser (λ = 630 nm, P = 5 mW) with two doses: 1 J/cm^2^ and 4 J/cm^2^, and the procedure was carried out under the influence of isoflurane. 5-ALA was applied on the wounds on the right hind paws, MB was applied on the left ones. The animals were divided according to the laser dose and photosensitizer into five groups: the control group, the LDPDT–5-ALA 1 J/cm^2^, LDPDT–MB 1 J/cm^2^, LDPDT–5-ALA 4 J/cm^2^, and LDPDT–MB 4 J/cm^2^. LDPDT procedure was repeated once immediately after wound formation.

### 2.3. Optical Coherence Tomography Protocol 

The experiments were carried out using optical coherence tomography (OCT) on the GANYMEDE−II system (Thorlabs, USA) with the basic scanning module OCTG-900. It is possible to obtain information on the optical characteristics, morphology, and elastic properties of biological tissues using OCT. The GANYMEDE-II system uses a superluminescent diode with an operating wavelength of 930 ± 50 nm. The superluminescent diode allows one to reach a signal penetration depth up to 2.9 mm with an axial resolution of up to 6.0 microns (air/tissue). The width of the spectral band was 100 nm. Figure 2 shows an example of placing a mouse on the substrate of OCT. As a result, B-scans were obtained—two-dimensional images. Data processing was carried out using ThorImageOCT 5.0.1., with the following parameters: size 2469 × 675 pixel, FOV 4.66 × 1.94 mm, and pixel size 1.89 × 2.88 µm, with 20 frames. The experiment was repeated with a 30° rotation around the previous position.

### 2.4. Statistical Analysis

The OCT data were exported using the ThorImageOCT 5.0.1 program to files. txt. Statistical analysis and data processing were carried out in Python 3.6 using libraries (numpy, scipy, matplotlib). All calculated parameters were expressed as the mean ± standard deviation. The Pearson test was applied to assess the level of statistically significant differences among groups under study. Statistical power was used at 0.95 and 0.99. The *p*-values were calculated for all groups on all days.

## 3. Results

### 3.1. OCT Imaging

Figure 3a shows a photo of healthy skin, and an OCT B-scan for the area, marked with a red arrow, is shown in Figure 3b. The B-scan data were normalized so that the stratum corneum, corresponding to the region with the highest pixel intensity, was located at the top of the image. After normalization, the signal intensity was calculated at different tissue depths from 0 to 0.8 mm (A-scan) and visualized as shown in Figure 4. The maximum intensity values were at a depth of 0 to 2–4 µm, which corresponded to the stratum corneum, and then the signal intensity gradually decreased to ~20 µm, which corresponded to the epidermis. The dermis starts from 30–40 µm, which was accompanied by a decrease in intensity to the minimum values at a depth of 0.8 mm.

Photos of the observation area and B-scans of the wound at different time points on days 1, 3, 7, and 14 for the control (without LDPDT) are shown in Figure 5.

The wound healed typically without pathologies, and the injury was close to healing on day 14. The difference in signal intensities at different stages of wound healing on different days is shown in Figure 6. On day 1 after the wound forming procedure, the signal had a low intensity, while the signal of the dermis started decreasing from 0.3 µm, so the signal from 0 to 0.3 corresponded to the formed wound scab. The signal intensity increased on days 3 and 7. On day 14, the signal intensity values were close to healthy skin.

OCT images for the LDPDT-5-ALA groups are shown for a laser dose of 1 J/cm^2^ in Figure 7 and 4 J/cm^2^ in Figure 8. The intensity signal for LDPDT-5-ALA 4 J/cm^2^ had the same behavior as the control. On day 1 for LDPDT-5-ALA 1 J/cm^2^, the attenuation signal corresponding to the dermis started from ~0.3 mm. Similar to the control, the signal intensity increased on days 3 and 7, and on day 14, the signal intensity values were close to healthy skin, as shown in Figure 9a. In the same way for LDPDT-5-ALA 4 J/cm^2^, the intensity signal started decreasing from ~0.2 mm on day 1. On days 3 and 7, the signal intensity values increased to close to the value of healthy skin on day 14 more than the LDPDT-5-ALA 1 J/cm^2^ group, as shown in Figure 9b.

Measurements were similarly carried out for the MB photosensitizer with two laser doses of 1 J/cm^2^ (Figure 10) and 4 J/cm^2^ (Figure 11). The intensity signal for LDPDT-MB on day 14 was close to the values for healthy skin for different exposure doses, as shown in Figure 12.

### 3.2. Quantitative Comparison of the Spatial Proximity of the OCT Signal Intensity

For a quantitative comparison of spatial profiles, we used the curve proximity factor (CPF) *S*, similar to the Pearson’s correlation coefficient, to compare healthy skin and wound curves in all days to all groups [47]:(1)$$S=\frac{\sum_{i}\left|X_{i}-Y_{i}\right|}{\frac{1}{2}\sum_{i}\left|X_{i}+Y_{i}\right|},$$
where *X_i_*, *Y_i_* is the intensity of the OCT signal for a definite depth from the wound and healthy skin, respectively. The higher the CPF value, the closer the wound state to healthy skin. The 0.01 and 0.05 significance levels were used to assess the statistical differences between the wound and healthy skin groups. The CPF values for all groups are shown in Table 1.

The CPF (1) was also used to estimate the effectiveness of different laser doses for each photosensitizer on day 14. The CPF values calculated for the OCT signal intensity curves corresponding to 4 J/cm^2^ and 1 J/cm^2^ for 5-ALA (Figure 9) and MB (Figure 12) are shown in Table 2. These quantitative estimations demonstrated a “proximity” between the curves corresponding to 4 J/cm^2^ laser dose and the curves corresponding to the 1 J/cm^2^ laser dose for the same photosensitizer.

## 4. Discussion

The proposed method of wound state quantitative evaluation is based on the OCT visualization of tissue structure transformation. The averaged scatter A-line intensity profile obtained from the horizontal rectangle in the OCT B-scan image of healthy skin is shown in Figure 4. Three areas are highlighted in the figure, representing changes in the attenuation coefficient. The red (S_1_), green (S_2_), and blue (S_3_) lines correspond to the beginning of the stratum corneum, the end of the epidermis, and the beginning of the dermis, respectively. After inflicting a wound in the first days, there are no surface layers of the skin (horny, epidermis); instead, a scab forms on the surface. These changes in the skin are visible on A-scans. Over time, the skin recovers, and on A-scans, we can see the appearance of areas characteristic of the epidermis’s end and the dermis’s beginning. These changes are reflected in the OCT signal attenuation curve (see Figure 6, Figure 8, Figure 9 and Figure 12).

For wound state quantitative estimation, we used the curve proximity factor, introduced by us earlier [47]. According to Table 1, in the control group, the CPF mean value on day 1 was about 0.053, and decreased to 0.021 on day 14, while in the LDPDT groups, the mean values of this coefficient on day 14 ranged from 0.017 to 0.021. The Pearson test demonstrated that for LDPDT groups on day 14, *p*-value did not exceed 0.01, while for the control group, this value was equal to 0.03. Therefore, for all LPDT groups, the wound state had no statistically significant difference compared to healthy skin for the used statistical power levels.

The CPF value for LDPDT groups for the 4 J/cm^2^ laser dose was smaller than the LDPDT groups for 1 J/cm^2^. We also calculated the CPF values for the OCT signal intensity curves corresponding to 4 J/cm^2^ and 1 J/cm^2^ for 5-ALA (the first column in Table 1) and MB (the second column in Table 2) and conducted a Pearson test to check the statistical significance of these differences. *p*-value was shown to be larger for 5-ALA. 

Therefore, after comparing the CPF parameter for five groups: control, LDPDT 5-ALA 4 J/cm^2^, LDPDT 5-ALA 1 J/cm^2^, LDPDT-MB 4 J/cm^2,^ and LDPDT-MB 1 J/cm^2^, it was concluded that the laser dose of 4 J/cm^2^ for LDPDT 5-ALA was definitely better compared to 1 J/cm^2^ and probably better for LDPDT MB. It should be noted that the conclusion depends on the volume and quality of the dataset.

A possible reason for the 4 J/cm^2^ dose preference relative to the 1 J/cm^2^ dose is as follows. According to previous works [33,34,35], when low-level laser light is applied and a dose is too low, no tissue response will occur. If too a high dose is applied, it can inhibit a tissue response. It has been seen in studies of wound healing where too low a dose did not have an impact, and too high a dose (above 5 J/cm^2^) prolonged wound healing while the optimal dose resulted in faster healing. In this interval, according to the Arndt–Schultz curve, the larger dose causes a stronger biological effect. 

In any case, LDPDT allows for accelerating of the wound healing process, which is consistent with the literature data [32,33,34,48,49].

## 5. Conclusions

In this paper, a study of quantitative in vivo evaluation of wounds using OCT during the wound healing process was carried out. 5-ALA and MB were used as photosensitizers for LDPDT, with two laser doses of 1 and 4 J/cm^2^.

An approach to quantitative estimation of wound state based on the CPF, Equation (1) [47] was proposed. The method was used to quantify the effectiveness of LDPDT to accelerate the wound healing process. CPF parameter estimation allowed us to compare LDPDT regimes quantitatively and to obtain objective arguments about the superiority of one regime over another.

Therefore, the proposed CPF parameter, estimated from OCT data, has demonstrated its feasibility for the quantitative estimation of the human wound state during healing. This approach is noninvasive, simple in implementation, and suitable for continuous monitoring throughout the wound healing process and sufficient resolution to assess both anatomy and pathology. It makes it a promising technique for applications in wound healing and the evaluation of novel therapeutics.

Another approach, which was proven to be effective in monitoring wound healing is two-photon microscopy. Previously, our group analyzed the two-photon microscopy images of the wound healing process and succeeded in quantitatively assessing the state of the wound and studying the effect of low-dose photodynamic therapy using the techniques of two-photon microscopy. The results of this study are completely consistent with the results obtained earlier [50].

## Figures and Tables

**Figure 1 pharmaceutics-14-00399-f001:**
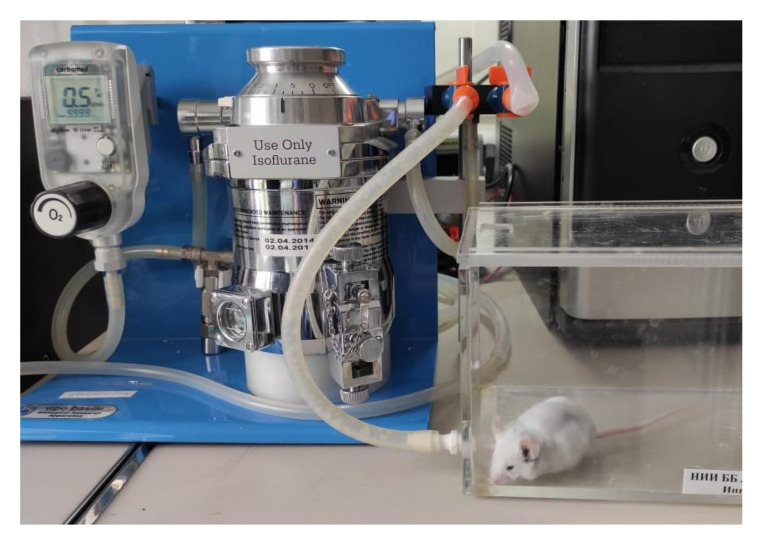
Anesthetized mice by isoflurane using the Ugo Basile gas anesthesia system.

**Figure 2 pharmaceutics-14-00399-f002:**
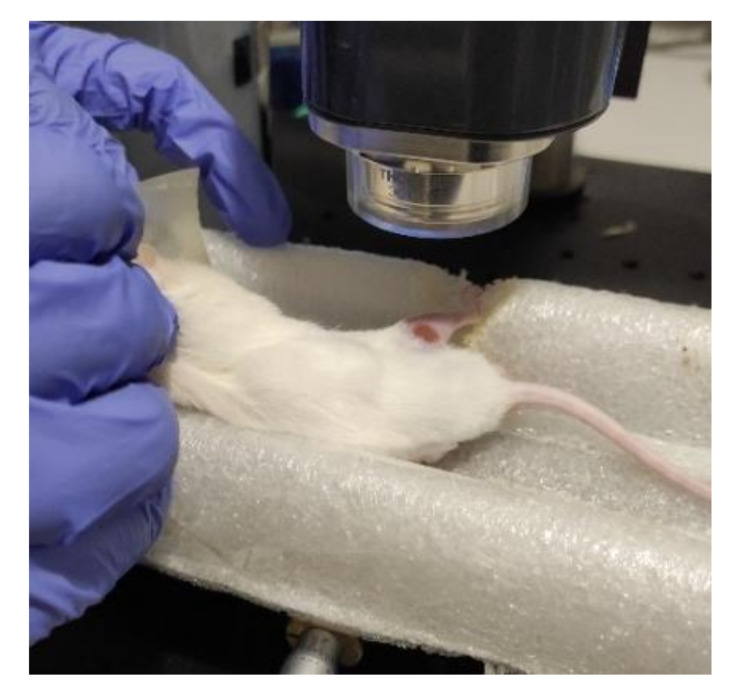
The mouse positioning during OCT imaging.

**Figure 3 pharmaceutics-14-00399-f003:**
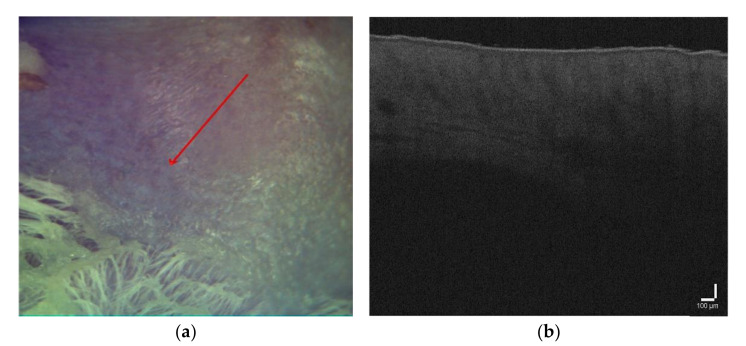
(**a**) Visual observation for healthy skin, (**b**) OCT imaging.

**Figure 4 pharmaceutics-14-00399-f004:**
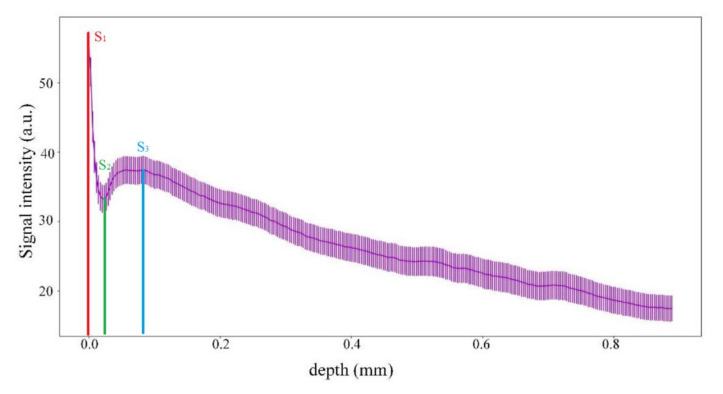
Dependence of signal intensity on depth for healthy skin: S_1_ start point of the stratum corneum; S_2_—the endpoint of epidermis; S_3_—start point of the dermis.

**Figure 5 pharmaceutics-14-00399-f005:**
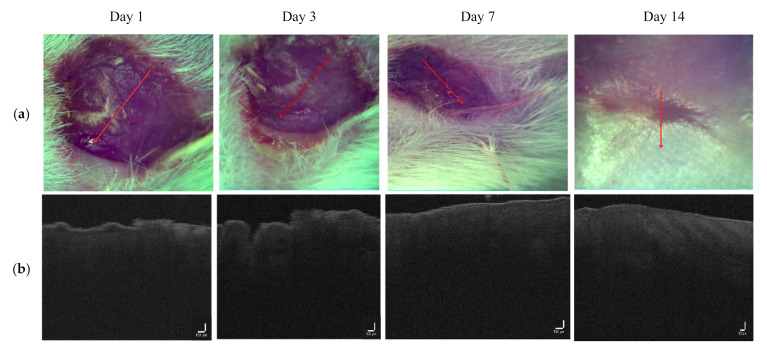
(**a**) Visual observation and (**b**) the corresponding B-scans for the control group.

**Figure 6 pharmaceutics-14-00399-f006:**
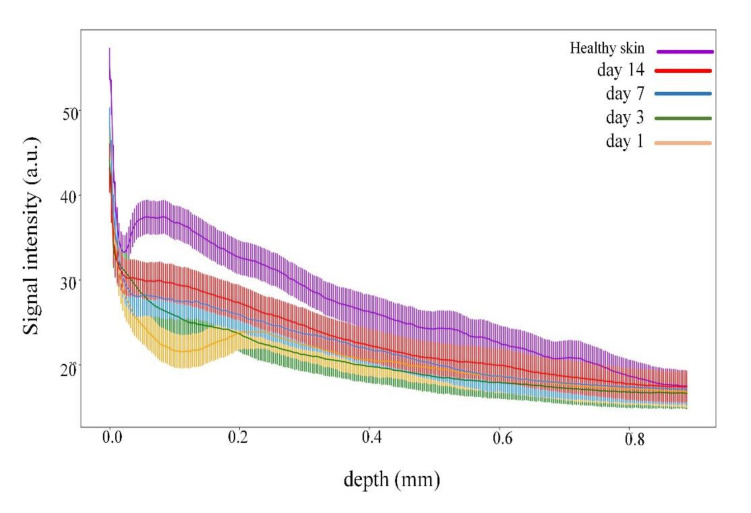
Dependence of signal intensity on depth for the control group during wound healing.

**Figure 7 pharmaceutics-14-00399-f007:**
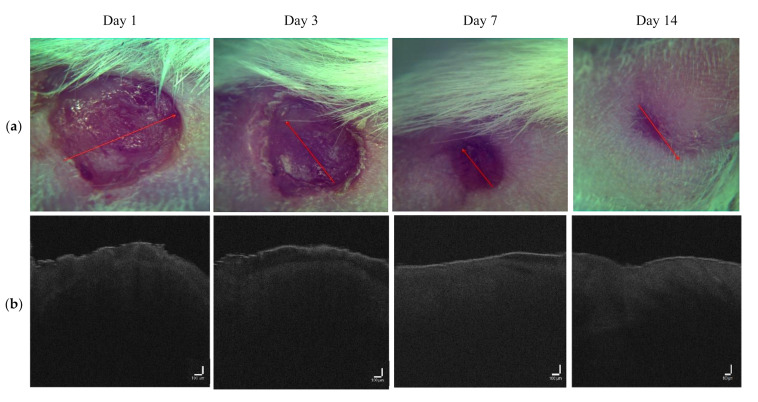
(**a**) Visual observation and (**b**) the corresponding B-scans for the LDPDT-5-ALA 1 J/cm^2^ group.

**Figure 8 pharmaceutics-14-00399-f008:**
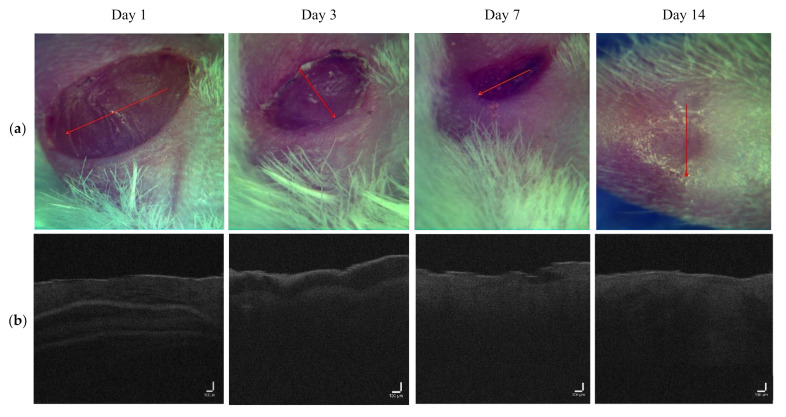
(**a**) Visual observation and (**b**) the corresponding B-scans for the LDPDT-5-ALA 4 J/cm^2^ group.

**Figure 9 pharmaceutics-14-00399-f009:**
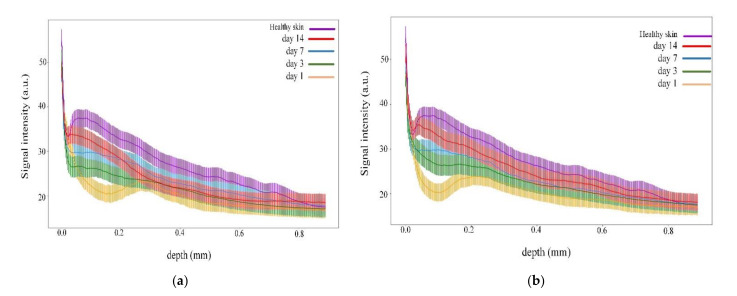
Dependence of signal intensity on depth for the (**a**) LDPDT-5-ALA 1 J/cm^2^ group and (**b**) LDPDT-5-ALA 4 J/cm^2^ group.

**Figure 10 pharmaceutics-14-00399-f010:**
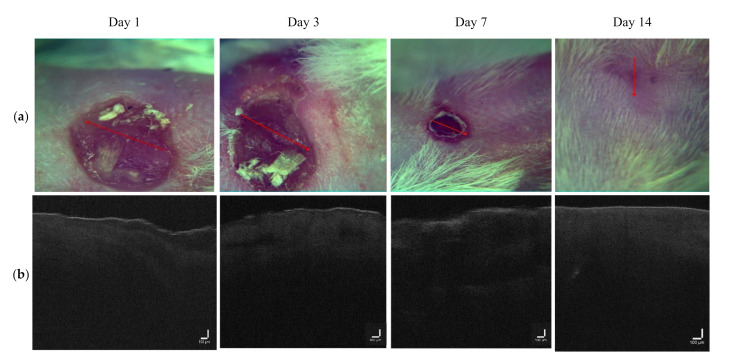
(**a**) Visual observation and (**b**) the corresponding B-scans for the LDPDT-MB 1 J/cm^2^ group.

**Figure 11 pharmaceutics-14-00399-f011:**
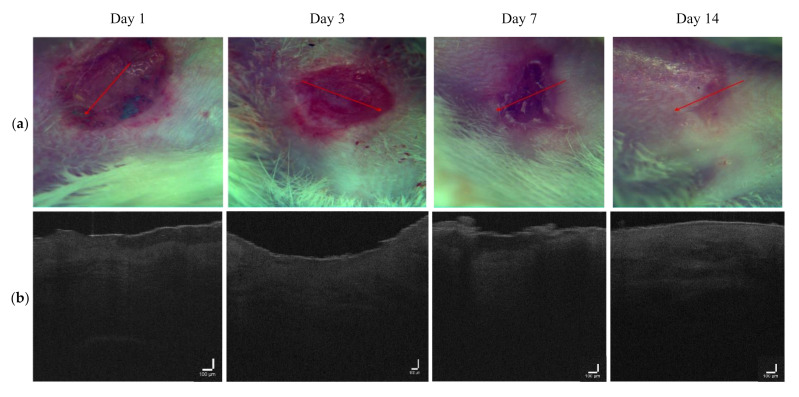
(**a**) Visual observation and (**b**) the corresponding B-scans for the LDPDT-MB 4 J/cm^2^ group.

**Figure 12 pharmaceutics-14-00399-f012:**
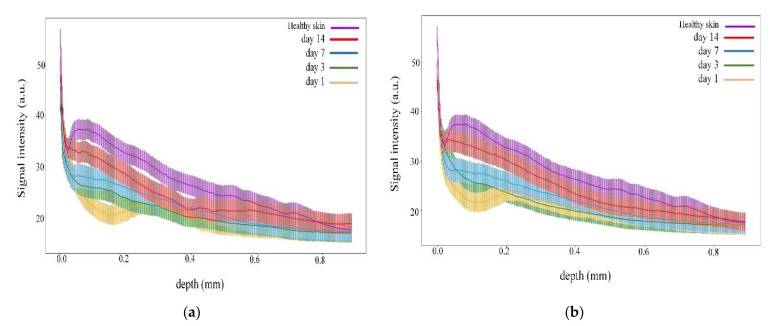
Dependence of signal intensity on depth for the (**a**) LDPDT-MB 1 J/cm^2^ group and (**b**) LDPDT-MB 4 J/cm^2^ group.

**Table 1 pharmaceutics-14-00399-t001:** CPF values for the studied groups (mean ± standard deviation).

*n* = 5	Control	LDPDT–5-ALA 1 J/cm^2^	LDPDT–5-ALA 4 J/cm^2^	LDPDT–MB 1 J/cm^2^	LDPDT–MB 4 J/cm^2^
Day 1	(0.0528 ± 0.0084) **	(0.0516 ± 0.0072) **	(0.0517 ± 0.0045) **	(0.0494 ± 0.0083) **	(0.0539 ± 0.0064) **
Day 3	(0.0404 ± 0.0141) **	(0.0504 ± 0.0106) *	(0.0480 ± 0.0087) **	(0.0456 ± 0.0169) *	(0.0423 ± 0.0134) *
Day 7	(0.0316 ± 0.0089) *	(0.0379 ± 0.0124) *	(0.0315 ± 0.0111) *	(0.0361 ± 0.0084) *	(0.0305 ± 0.0073) *
Day 14	(0.0214 ± 0.0076) *	(0.0213 ± 0.0075)	(0.0187 ± 0.0213)	(0.0201 ± 0.0054)	(0.0174 ± 0.0051)

* *p* > 0.01,** *p* > 0.05.

**Table 2 pharmaceutics-14-00399-t002:** CPF values for the studied groups (mean ± standard deviation).

*n* = 5	LDPDT–5-ALA	LDPDT–MB
Day 14	(0.0115 ± 0.0019) **	(0.0142 ± 0.0011) *

* *p* > 0.005, ** *p* > 0.01.

## Data Availability

The data presented in this study are available on request from the corresponding author.

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
