# Peer review of "The In Vivo Quantitative Assessment of the Effectiveness of Low-Dose Photodynamic Therapy on Wound Healing Using Optical Coherence Tomography"

_pharmaceutics, 2022, doi:10.3390/pharmaceutics14020399_

Round 1
Reviewer 1 Report
An interesting original study using optical coherence tomography to evaluate the quickness of wound healing after the use of various types of photodynamic therapies, showing that higher powers and methylene blue are slightly more effective in favoring wound healing.
I have some queries before accepting the paper for publication:
Statistical significance, if I understood correctly, only applies to confront the control group to each of the subgroups, showing significant differences. Have you compared also the subgroups? Is the difference in CPF values significant?
I did not find very clear the part of statistical analysis....the program versions should be better specified.
The discussion should be broadened, and the authors should try to explain the results they obtained in this section (why methylene blue and 4 J/cm2 worked better?)
Thank You
Author Response
-
-
- Statistical significance, if I understood correctly, only applies to confront the control group to each of the subgroups, showing significant differences. Have you compared also the subgroups? Is the difference in CPF values significant?
In this study, we compared each group (including the control group) with a healthy tissue group to show the deviations from healthy tissue.
In the process of wound healing, we noticed that the CPF values differences became smaller compared to healthy tissues. Also, statistical difference was estimated among groups, and days of the experiment using p-value significance levels are 0.01 and 0.05. Finally, we compared the subgroups and identified the best dose and photosensitizer. Taking into account your questions and comments, we rewrote and supplemented the protocol and results. Also, in section 3.2 we added the CPF values to compare between the laser doses 4J/cm2 and 1J/cm2 in Table 2. To illustrate the significance in the CPF values, we used the Pearson test (see p-values at the bottom of Tables 1, 2).
- I did not find very clear the part of statistical analysis....the program versions should be better specified.
The information and details about the program version and library were added to sections 2.3 and 2.4.
- The discussion should be broadened, and the authors should try to explain the results they obtained in this section (why methylene blue and 4 J/cm2 worked better?)
In Table1, we added p-values that characterize the significance of the CPF values variations relatively to healthy skin group in the data obtained. Also, we added Table 2 with CPF values calculated for a definite LDPDT group on day 14 and different laser doses to estimate the better laser dose for each photosensitizer. It was explained how and why conclusions were made about the used laser radiation doses and photosensitizers. A possible reason for 4 J/cm2 dose preference relatively 1 J/cm2 is also commented in the Discussion section.
Also, for better understanding, we changed Figure 4; the links between the A-stack and the layers of the skin have been shown.
All these new results were additionally commented in the Discussion section.
-
Reviewer 2 Report
Please see the comments are below
- Did author attempt to check any other photosensitizers other than 5-ALA and MB?
- Did author attempted to use laser doses other than 1 and 4 J/cm2.
- Why did author choose laser dose 1 and 4 J/cm2?
- Why the 4J/ cm2 is better than 1J/cm2? Give the reasons.
- Did author attempt to check generation of ROS?
Author Response
- Did authors attempt to check any other photosensitizers other than 5-ALA and MB?
In our first experiments in this field of study, we used only the photosensitizers: 5-ALA and MB. When choosing photosensitizers, we were guided by a literature review previously conducted in the framework of this study. Literature excerpts were added to the introduction.
- Did author attempted to use laser doses other than 1 and 4 J/cm2.
In our first experiments in this field of study, we used only the laser doses: 1 and 4 J/cm2. When choosing the laser doses, we were guided by a literature review previously conducted in the framework of this study. Literature excerpts were added to the introduction.
- Why did author choose laser dose 1 and 4 J/cm2?
After reviewing Literary sources and previous experiences in this field and according to the Arndt-Schultz Law, the laser doses 1 and 4 J/cm2 were chosen, more information and details about the laser doses were added to the introduction.
- Why the 4 J/cm2 is better than 1 J/cm2? Give the reasons.
For a wound state quantitative estimation, we used the curve proximity factor. According to table 1, CPF values were calculated for all groups on all days. In Table 2, СPF values for OCT signal intensities curves corresponding to 4 J/cm2 and 1 J/cm2 for both photosensitizers: 5-ALA and MB. After comparing the CPF parameter for the studied groups, it was concluded that the laser dose of 4 J/cm2 is definitely better compared to 1 J/cm2. This result was confirmed by the Pearson test (see p-values at the bottom of Table 2). The results were explained in detail in the Discussion section. A possible reason for 4 J/cm2 dose preference relatively 1 J/cm2 is also commented in the Discussion section.
- Did authors attempt to check the generation of ROS?
Thank you very much for the question. Many researchers [1] have already studied and demonstrated that MB is an antioxidant. Results about ROS for 5-ALA also were published [2]. In this work, we carried out only in vivo measurements since we studied the dynamics of the wound healing process, and because of small samples, it was essential to observe the same wounds to reduce scatter. The measurement of reactive oxygen species in tissues in vivo is rather complicated. We have a technical ability to conduct ROS chemiluminescent analysis in vitro, but its adaptation for skin in-vivo analysis is too complex.
[1] Xiong, ZM., O’Donovan, M., Sun, L. et al. Anti-Aging Potentials of Methylene Blue for Human Skin Longevity. Sci Rep 7, 2475 (2017). https://doi.org/10.1038/s41598-017-02419-3
[2]Morimoto, K.; Ozawa, T.; Awazu, K.; Ito, N.; Honda, N.; Matsumoto, S.; Tsuruta, D. Photodynamic Therapy Using Systemic Administration of 5-Aminolevulinic Acid and a 410-Nm Wavelength Light-Emitting Diode for Methicillin-Resistant Staphylococcus Aureus-Infected Ulcers in Mice. PLoS ONE 2014, 9, e105173, doi:10.1371/journal.pone.0105173.
Round 2
Reviewer 1 Report
The paper improved after revisions. it is now eligible to be published.